# Developing and Implementing a Lean Performance Indicator: Overall Process Effectiveness to Measure the Effectiveness in an Operation Process

Lisbeth del Carmen Ng Corrales [1,2,*], María Pilar Lambán [1], Paula Morella [1], Jesús Royo [1], Juan Carlos Sánchez Catalán [3] and Mario Enrique Hernandez Korner [1,2]

[1] Design and Manufacturing Engineering Department, Universidad de Zaragoza, 50018 Zaragoza, Spain; plamban@unizar.es (M.P.L.); pmorella@unizar.es (P.M.); jaroyo@unizar.es (J.R.); mario.hernandez2@utp.ac.pa (M.E.H.K.)

[2] Department of Industrial Engineering, Universidad Tecnológica de Panamá, Ciudad de Panama 0819-07289, Panama

[3] Smart Systems, Tecnalia, Basque Research and Technology Alliance (BRTA), 20009 Donostia-San Sebastian, Spain; jcarlos.sanchez@tecnalia.com

[*] Correspondence: lisbeth.ng@utp.ac.pa

**Abstract:** The purpose of this paper is to build up and implement a framework of a lean performance indicator with collaborative participation. A new indicator derived from OEE is presented, overall process effectiveness (OPE), which measures the effectiveness of an operation process. The action research (AR) methodology was used; collaborative work was done between researchers and management team participation. The framework was developed with the researchers' and practitioners' experiences, and the data was collected and analyzed; some improvements were applied and finally, a critical reflection of the process was done. This new metric contributes to measuring the unloading process, identifying losses, and generating continuous improvement plans tailored to organizational needs, increasing their market competitiveness and reducing the non-value-add activities. The OEE framework is implemented in a new domain, opening a new line of research applied to logistic process performance. This framework contributes to recording and measuring the data of one unloading area and could be extrapolated to other domains for lean performance. It was possible to generate and validate knowledge applied in the field. This study makes collaborative participation providing an effectiveness indicator that helps the managerial team to make better decisions through AR methodology.

**Keywords:** OEE; KPIs; action research; lean; operation process; performance

## 1. Introduction

To remain competitive, companies must constantly review the performance of their operations and production processes. To improve the operation process, it is not always necessary for investments in technology, equipment, and facilities, but instead, the increase of the efficiency could be made with small improvements associated with process sustainability [1]. It is useful for the company to find the appropriate tools to measure the efficiency and productivity of a transport process to eliminate or reduce non-value-added activities [2]. Previous research in transport value has established through interviews with practitioners that there are two types of value-added activities, transportation, and unloading/loading activities [3]. This study's novelty is that it develops and implements an adaptation of the Overall Equipment Effectiveness (OEE) indicator to measure the effectiveness of a loading or unloading process. In this research, this indicator is adapted and implemented in a logistic process to evaluate and measure every step of the selected process. The OEE has been used as a lean metric to identify and eliminate process losses [4,5].

OEE indicator is used to measure the effectiveness level, identifying losses and allowing with a later analysis to identify the non-value-adding activities. OEE is an indicator of effectiveness, although it is often confused with efficiency. However, effectiveness indicates the objectives achieved and efficiency relates to the use of resources and the achievements attained [6]. A considerable amount of research has been published in recent years [7] concerning the OEE, such as case studies, implementations, original OEE slightly modified, and new adaptations to other areas. The OEE has been adapted to various domains depending on industry need, such as line manufacturing [8], transport [9], mining equipment [10], and assembly tasks [11]. Authors proposed areas in which OEE could be applicable, in the service sector and logistics processes, such as goods reception or performing selection in a warehouse [1,12,13]. This indicator is appreciated by managers because it is an overall metric, simple and clear, which identifies areas of potential improvement.

The purpose of this paper is to build up and implement a framework of a lean performance indicator with collaborative participation. The implementation of this framework makes it possible to establish a procedure to record, measure, and study the process of lateral unloading trucks. An OEE framework adapted to the logistic field will be implemented and the identification of the losses that reduce the effectiveness of the process is intended. To achieve this objective, the OEE indicator traditionally used in production will be adapted to the logistic area to measure the effectiveness of the unloading truck process. A new metric, named Overall Process Effectiveness (OPE), is developed to quantify the effectiveness of the logistic process, identifying the hidden losses that most affect the unloading truck process.

To fulfill the goal of this study, action research (AR) methodology was selected. The AR used in this study seeks to solve an organizational issue while building up scientific knowledge. AR is a collaborative work of research with participation in the field. The term was used in works of the German psychologist Kurt Lewis [14] that considers the impact linked between research and action to create change in the organizational process. The AR must be researched in action, participative, a sequence of events, concurrent with action, and provide an approach to problem-solving [15].

The paper is organized as follows: in Section 2 described and explained the performance measurement systems, the origin of the OEE concept, and the six big losses. Section 3 the proposed methodology and case description are displayed. The framework to be implemented is developed in Section 4 and in the next section, the analysis of the results is presented. Finally, Sections 6 and 7 show the discussion and conclusion of the research, respectively.

## 2. Theoretical Underpinning

### 2.1. Performance Measurement System

Companies have a great interest in performance measurement systems as they provide insight into the performance of a specific activity. Proper implementation and use of the performance measurement systems provide a clearer picture of the performance of the company's operational processes.

A performance measurement system can guide and influence the decision-making process. Since the end of the 1980s, performance measurement systems have become a relevant issue for scholars and practitioners [16]. Over the last years, companies increase their competitiveness by implementing performance measurement systems as platforms of continuous improvements in organizational processes [17]. The performance measurement systems can be defined as the process of quantifying the efficiency and effectiveness of a certain activity with a set of metrics. It can be examined at three different levels: individual performance measures, set of performance measures, and environment within which the performances measurement system operates [18].

Over the years, companies have implemented performance measurement systems in different areas and sectors such as finance, process, human resources, and service. These systems facilitate monitoring the efficiency of time and resources, helping to set

realistic goals, objectives, and improvements. A good performance measurement system provides the company feedback on job performance, a guide for defining new and clear goals, and generating increases in the profit margins. Caplice and Sheffi in 1995 [19] state that a good logistics performance measurement system should be comprehensive, causally oriented, vertically integrated, horizontally integrated, internally comparable, and useful. From a strategic point of view, performance measurement is not only the manager's responsibility, but field operators must also be involved in the implementation. A logistics performance measurement system must include multiple goals to be viewed within the organizational performance. Caplice and Sheffi [20] argued that must be considered in a logistics performance indicator, validity, robustness, usefulness, integration, economy, compatibility, level of detail, and behavior soundness.

According to the literature [21,22], the three steps to develop a performance measurement system are:

a.   Design: this phase involves planning what and how to measure. It also identifies the key objectives to be achieved and the areas involved in the data collection.
b.   Implementation: the systems and procedures are put into action. In this phase, the output of the results is verified, and the system is refined if necessary.
c.   Use: the results are reviewed. The process or resource understudy is checked to see if it is being effective and under the company's goals.

To quantify organizational and individual performance, Key Performance Indicator (KPI) can enable an objective measurement, based on organizational goals. KPIs monitor critical business activities and provide information to increase organizational performance [23]. Industry 4.0 has encouraged digital transformation in the organization. Digital transformation allows having reliable, truthful, and real-time information to feed the management information system to make better decisions [7,24], though digital data is possible to enhance a data-driven and more intelligent approach toward the continuous improvement process [25]. OEE is a KPI widely accepted as a tool well implemented in lean manufacturing to monitor the performance of a process.

Traditionally in general practice, individual performance was measured from utilization, productivity, and effectiveness metrics [20]. For this research, a multi-dimensional perspective is considered for a new metric [26]. The proposed indicator is the OEE, a key performance indicator used to measure the availability, performance, and quality in a process. Moreover, the indicator allowed us to identify productivity losses during the process for continuous improvements.

### 2.2. Overall Equipment Effectiveness

Initially, the OEE was a quantitative metric of the total productive maintenance launched by Nakajima in 1988. Muchiri and Pintelon in 2008 [6] defined OEE as a tool of performance that measures any losses in production and identifies the areas of process improvement. This metric measures the availability, performance, and quality of individual equipment showing the productivity in the manufacturing operations. OEE is the results in a percentage of the multiplication of these three parameters:

$$OEE\ (\%) = Availability \times Performance \times Quality$$

Availability measures the productive time of the machine or equipment when scheduled or available. This parameter is affected when there are losses due to equipment failure or setup and adjustment. Some activities are not included in the available time like the planned downtime such as planned maintenance, labor meeting, among others. If the planned downtime is considered in the production time, the result would be lower, but the true availability would be shown [27].

Performance refers to the real production versus the productive capacity of the machine. It can be also measured by the actual time cycle time against the ideal cycle time. Performance is affected by speed losses due to idling and minor stoppage or reduced speed.

Quality focuses on the quantity of product that meets the standards versus the total production. Quality may be affected by-products that do not meet established standards, damaged products, or scrap.

The OEE range has been the subject of discussion over the years by different authors. Ref. [28] considers that there is no optimal number of the indicator since different criteria depend on the industry where it is applied. Ref. [29] proposes 85% as the ideal OEE value. Worldwide studies indicate that the average OEE rate in manufacturing plants is 60% [30]. Other authors differ in the indicator range between 30 to 80%, with being the 50% the more realistic value [27].

Several authors define the OEE depending on the application and situation. Table 1 presents a comparison between [29,31] variables definition. Nakajima defined the variables depending on equipment performance on the other hand Braglia established the variable based on the labor performance.

**Table 1.** Variable definition OEE from Nakajima and OEE adaptation from Braglia.

| Variable | [29] | Variable | [31] |
|---|---|---|---|
| **Quality (Q)** | $\frac{input - volume\ of\ quality\ defects}{input}$ | Overall workplace effectiveness (OWE) | $\frac{value\ time}{operating\ time}$ |
| **Performance (P)** | $\frac{ideal\ cycle\ time * output}{operating\ time}$ | Management effectiveness (ME) | $\frac{operating\ time}{effective\ available\ time}$ |
| **Availability (A)** | $\frac{loading\ time - downtime}{loading\ time}$ | Worker availability (WA) | $\frac{effective\ available\ time}{net\ available\ time}$ |
| **OEE** | $A \times P \times Q$ | ROLE | $OWE \times ME \times WA$ |

Authors like [32] consider that the gains in OEE, while important and ongoing, are insufficient because no machine is isolated. OEE is widely accepted as a quantitative tool, but it is limited to the productivity behavior of individual equipment [33]. Due to the needs presented in the industry, the OEE has led to modification and enlargement of the original OEE tool to fit a broader perspective [6]. OEE is an effective management metric that has adapted and gained importance in different industries becoming one of the most important measures in the modern industry [34,35].

OEE is traditionally used to monitor production performances, but it can also be used as a metric for process improvement activities in other contexts [36]. In the production context, OEE has been used to measure the productivity and performance of a line manufacturing system by detecting and quantifying the line's critical points [8,37,38]. It has also been possible to measure losses of resources associated with human, material, and machine factors [39–41]. In the mining sector, the OEE has been modified to measure the effectiveness of mining equipment such as draglines, shovels, and trucks [10,42]. Garcia-Arca et al. [1] improved efficiency in road transport and Ref. [9] optimized the effectiveness of urban freight transportation by adapting OEE. Concerned about the environment and raising awareness to contribute to a more sustainable world, Refs. [4,43] proposed modifications to the OEE, including measurement of sustainability and performance in the production system.

### 2.3. Six Big Losses

The OEE indicator is designed to identify losses that reduce the effectiveness, Ref. [29] define six large losses that affect the availability (downtime losses), performance (speed losses), and quality (quality losses) factors:

Losses that affect the availability:

- Equipment failure or breakdown of these losses are caused by major shutdowns producing losses time and quality losses.
- Setup and adjustment of these losses of time are caused by the change of requirements in production at the end of one item to another item.

Losses that affect the performance:

- Idling and minor stoppage losses when production is interrupted by a temporary malfunction or when a machine is idling
- Reduced speed losses refer to the difference between the equipment design speed and the actual operating speed.

Losses that affect the quality

- Reduced yield occurring from the start-up to stabilization
- Quality defects and reworks losses

Losses or disturbances could also be classified depending on how often they occur as chronic or sporadic [27,44]. Chronic disturbances are usually small, hidden, and complicated to identify because they appear as normal, whereas the sporadic are obvious, occur quickly, irregularly, and with dramatic effects.

### 3. Methodology and Unloading Process Description

For this study, AR methodology was followed to encourage the study in a logistics process. One of the aspects that have motivated the application of this methodology is to contribute with qualitative studies in the logistics sector [45]. Another advantage of using AR is that it allows addressing a gap in theory with an explorative approach [46]. AR is one form of a case of study in which a sequence of activities and an approach is used to solve a problem. These practices involve data gathering, reflection on the action as it is presented through the data, generating evidence from the data, and making claims to knowledge based on conclusions drawn from validated evidence [47].

This paper attempts to solve an organizational issue and validate the application of OEE in a logistic process by applying AR to enrich the field's body of knowledge. We based our study on the AR cycle, and the phases of planning, acting, observing, and reflecting were included. Figure 1 presents the action research model and the phases applied in this study.

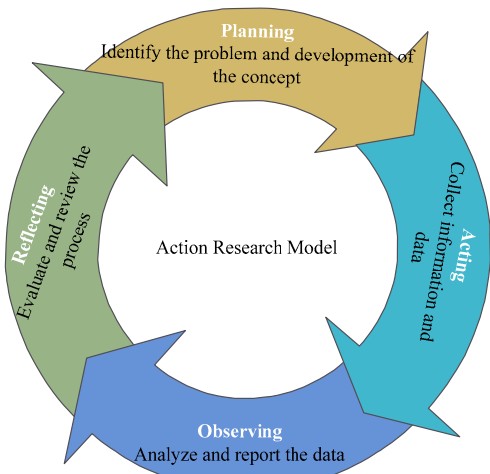

**Figure 1.** Action Research Model Phases used in this study.

According to the literature, three major categories, design, data collection, and data analysis aspects [48], make up a rigorous and relevant research approach. In the first category, research question, contribution in the practice and the science, motivation of the research, and the case description are reported subsequently. Although all the aspects require accompaniment by the company, this first required the experience of the researchers, since it was necessary to define the research questions, to know and understand the needs of the company and how the theoretical aspects of the indicator were going to be adapted and implemented in the chosen process. In the second category, the description, method, and access for data collection, the role of the team in charge, and the triangulation of experiences were considered during the action process. In team meetings (researchers and

practitioners) of reflection and discussion, the roles of each team member were defined and reliable systematic methods for data collection were implemented. Finally, the data were categorized and validated through the proposed indicator, the results were presented and after reflection, meetings with the whole team, the model was updated.

As established in the literature for the development of rigorous research, research questions are established, which we will answer using the AR methodology. This paper attended to answer these research questions:

RQ1. Can the new indicator measure the effectiveness of the lateral unloading truck process?

RQ2. Does the AR methodology provide solutions to the organizational issue?

RQ3. Has this study identified opportunities for improvements in the unloading process?

This methodology helps to create knowledge with collaborative participation, commonly in a specific and practical context to improve conditions and practices in the domain.

This research is carried out in one of the assembly plants of a multinational company in the automotive sector in Spain. The sub-department of material handling that belongs to the supply chain department, in search of improving the efficiency of its processes wants to implement a procedure to record, measure, and study the process of lateral unloading trucks. The project team is comprised of a researcher team, hereinafter researchers, and a management team. The researchers are the authors of this study specialized in the following areas:

- Experts in logistics processes;
- Experts in the OEE indicator applied in other engineering fields.

The management team is the practitioners been part of the company and consists of:

- Head of material handling department;
- Shifts supervisors;
- Shifts operators.

Through a reflection phase, with the professional experience and the theoretical knowledge of the researchers, the variables that influence the unloading process were defined and characterized. To broaden the theoretical knowledge of the performance measurement system, an indicator based on OEE was proposed for the logistics process.

This work was developed in four phases, where the researchers were presented and participated together with the company for the development and implementation of the indicator. Firstly, in the planning phase, the problem presented in the material handling sub-department was identified, a variation in unloading effectiveness was observed according to the work shift based on the number of trucks unloaded per shift. This phase had a duration of one month in which the researchers gained knowledge from the process at the company. During this phase, the researchers met with the management team to understand and define the scope and parameters of the research. Subsequently, to define an indicator to measure and improve the process of unloading material from different suppliers, an effectiveness indicator named OPE will be developed to implement it in the following phases.

In the second phase, the data is obtained from two sources of information: the routes registration system (LST) and the unloading operator. New data collection forms were created to obtain additional information from the unloading operator. The data were collected for five months. For data reliability, the company provided access to the LST system and proposed data collection from field operators. Meetings and training on the new measuring instruments were held with the operators, in this phase, the researchers actively participated.

Once the data is collected, in a third phase the indicator is calculated, and the results were analyzed. The results were analyzed through different perspectives for the support and implementation of improvements. Several meetings were held between the researchers and the management team, in which feedback on the results was given, changes on data

acquisition were proposed, and process improvements were suggested. This research involved the collaborative participation of the company and the researchers to reach reflections from different points of view. Through these reflections we gain a wider scope, reducing bias, and by validating the analysis to increased scientific rigor, achieving the triangulation mentioned by some authors [48,49].

In the last phase, there is a further clarification of the initial situation, a meeting for self-reflecting takes place among researchers, and the management team considers defining the improvements in the process. Despite reflection being placed as the last phase of the RA, it was carried out in each of the phases of the development and implementation of the new indicator in the company.

*Unloading Process Description*

The area chosen for the study is the unloading area with the highest flow of trucks, where the material is received from approximately 150 suppliers, and in 1780 different references are handled. The company works 24 h a day divided into three shifts: morning shift (6:00–14:00), afternoon shift (14:00–22:00), and night shift (22:00–6:00). All shifts have four breaks (one 8-min break and three 9-min breaks) and an 18-min snack break. Depending on the circumstances and the number of work breaks that can be worked on, but by law, it is not possible to work during the snack break. The unloading area has a reception area (temporary storage) and two unloading tracks, allowing two trucks to be parked in the unloading position (Figure 2). The unloading process is performed by a single operator, which could increase the waiting time between trucks.

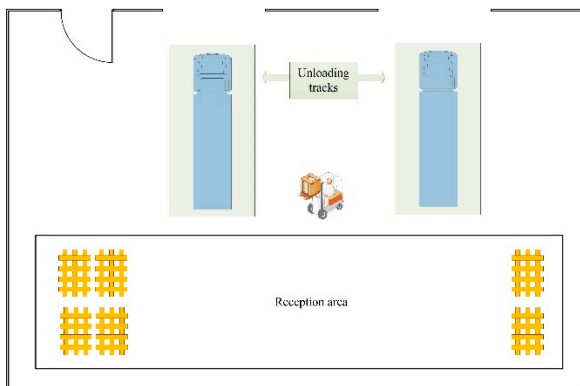

**Figure 2.** Physical distribution of the unloading area.

The process begins with the arrival of the truck at the external part of the unloading area, the driver gets off with the documentation corresponding to the transported material he is transporting. Then, the driver enters the warehouse, hands over the documents to the unloading operator who checks if the material belongs to that unloading section. If it does, he tells the driver to park the truck in the unloading track, otherwise, he is shown where to go. The operator records the following data in the document: route, time window, time of entry, time of exit, supplier, and registration.

Once the truck is on the unloading track, the driver gets off, puts on helmets and boots to start the process of preparing the truck to unload the material. The driver opens the side tarp of the truck, removes the side safety boards, and waits for the unloading operation. The operator begins the process by verifying that the material inside the truck is in good condition, if so, he proceeds to unload the containers. Otherwise, if the products contained are disordered in the truck, not loaded sideways, or in poor condition (wet, damaged, dumped, etc.), he calls the supervisor to verify the integrity of the container to proceed with the unloading process or asks the truck to leave. The operator uses a long-shovel forklift to unload the material and place it in the reception area, where it is temporarily stored and then placed in the warehouse. The material is stored in containers, the containers of the

same type are placed together. It is also considered that all the material transported in the same truck must be placed in the reception area as close as possible.

Once the unloading process is finished, the operator signs the document confirming, the driver proceeds to place the lateral safety boards, close the tarp, and leave the premises, leaving the unloading track for the next truck. Before continuing with the next truck, the operator must perform a verification of the products in the containers. The list of materials detailed in the documentation provided by the driver upon arrival must coincide with the material that was unloaded from the truck. If there is a difference, the operator takes note of it and notifies the supervisor to report the discrepancy. Figure 3 presents a flowchart of the unloading process.

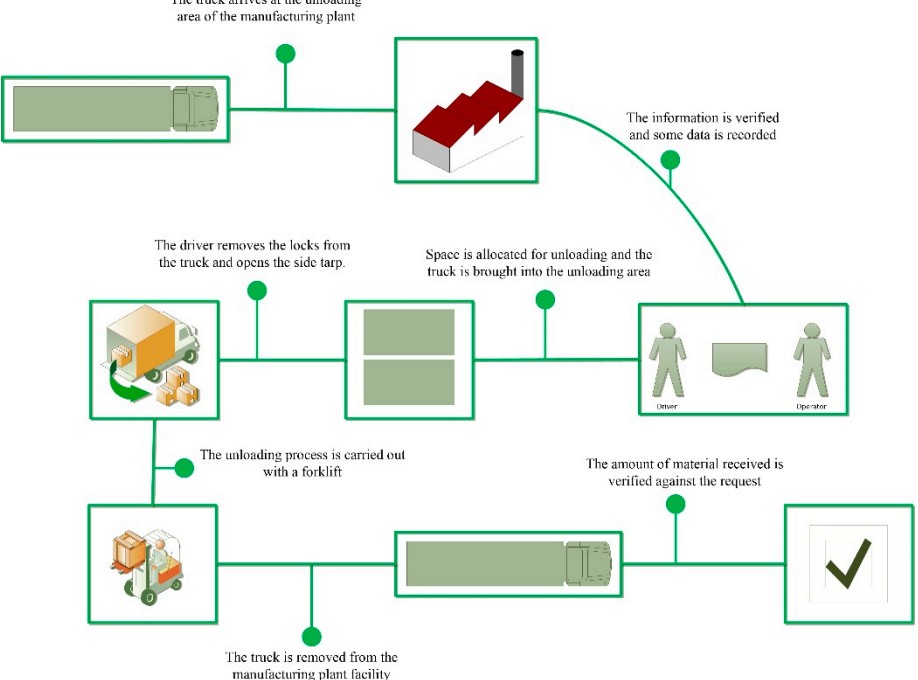

**Figure 3.** Unloading process flowchart.

The company has a delivery truck that travels within the manufacturing plant. This truck is used to transport the materials that were unloaded in another zone that does not correspond to it. The delivery truck has priority when it arrives at an unloading area and must be attended, once the operator has finished with the in-process truck unloading.

Once the researchers understood the unloading process, a framework was proposed that would meet the objective of this study. Through the framework, it was possible to find activities that do not add value and establish actions for continuous improvement. The OEE is an indicator widely used in manufacturing that looks forward to the effectiveness of the process [1]. The reason to choose this indicator is to adapt it to this research.

## 4. Developing the Framework

Through this framework, the company is oriented to reach a strategic objective to increase the overall efficiency in the production support activities. Based on the review of the existing OEE approaches, researchers proposed to the management team a new indicator OPE. The aim of this new indicator OPE is to measure the effectiveness of a logistic process. This framework considers that the effectiveness of the process is affected by internal and external factors. Internal factors are those situations that occur within the company and are managed and controlled internally in the organization. In contrast, external factors originated outside the company, depending on the participation or collaboration of

external agents to manage and control them. Both factors influence and affect positively or negatively the performance of the company.

The factors that were analyzed for the development of the indicator are availability, performance, quality, and punctuality. The availability measures the real total time the system is operating without time losses due to the lack of resources that prevent the process from running smoothly. The resources that affect the availability are the lack or failure of the forklift, lack of operator, among others.

$$Availability\ (A) = \frac{operating\ time}{total\ time} \times 100\% \tag{1}$$

$$Operating\ time = total\ time - downtime$$

$$Total\ time = workday - planned\ downtime$$

The workday can be calculated for the entire day or the work shift.

The performance measures the ratio of the ideal speed of the unloading and actual unloading speed. The performance is affected by losses that slow down the process.

$$Performance\ (P) = \frac{Ideal\ time}{Real\ time} \times 100\% \tag{2}$$

The ideal time is the historical average of the unloading time multiplied by the number of downloads made in the workday.

The quality rate measures the number of trucks that arrive with the material in satisfactory order so the process of lateral unloading can be carried out without delay.

$$Quality\ (Q) = \frac{Number\ of\ trucks\ as\ requested}{Total\ number\ of\ trucks} \tag{3}$$

Besides, the availability, performance, and quality, a new factor is added, related to the timely arrival of the truck. Punctuality is a ratio that indicates the portion of trucks that arrive in the assigned time window.

$$Punctuality\ (U) = \frac{Number\ of\ trucks\ arrive\ on\ time}{Total\ number\ of\ trucks} \tag{4}$$

The quality and punctuality factors are affected by internal and external situations, whereas the availability and performance factors are affected by situations entirely internal to the company. Table 2 shows a detailed list of losses classified by availability, performance, quality, and punctuality.

**Table 2.** Breakdown of losses by availability, performance, quality, and punctuality.

| Category | | Losses |
|---|---|---|
| Availability | Internal | Damage to the access door<br>Forklift discharge or breakdown<br>Lack of forklift |
| Performance | Internal | Occupational accident<br>The material drop during the unloading<br>Documentation check<br>Loading or unloading of the internal truck<br>Temporary storage area full |
| Quality | Internal | Truck for another uploading point |
| | External | Disordered material in the truck<br>Material not loaded sideways<br>The material is in poor condition (wet, damaged, dumped, etc.) |
| Punctuality | External | The truck does not arrive at the assigned time window |

This framework goes in pursuit to integrate important aspects of the process in a single measure of effectiveness (Figure 4). The perspectives to be measured with this framework are the availability of resources, the performance of resources, the quality of the process, and the added factor of punctuality. This last factor seeks to include a broader picture that affects the development of the process.

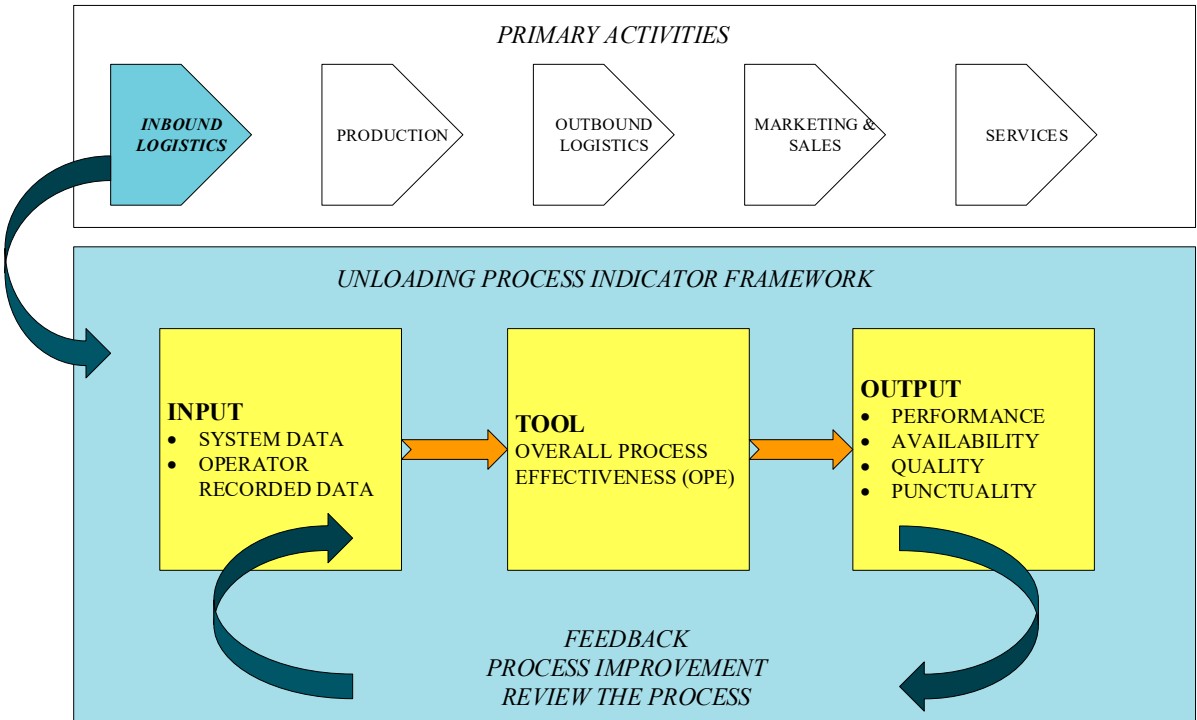

**Figure 4.** Unloading process indicator framework.

This new metric measures the effectiveness of the unloading process considering the four factors described above (see Equation (5)). It will be possible to identify and measure the losses that help to implement corrective actions to improve the effectiveness of the process.

$$OPE = A \times P \times Q \times U \tag{5}$$

The application of this integrated indicator can reveal through global OPE monitoring which of the four factors (availability, performance, quality, and punctuality) could be improved to increase the productivity of the process.

## 5. Results of the Framework Implementation

The results presented below are the product of several meetings between the researchers and the management team. The development of the research results is based on the acting and observing phases where the information is collected and analyzed. The proposed framework for measuring the performance of the unloading process through the OPE was presented to the management team. Meetings were held with the operators to indicate the process for template records to collect the data.

The information was worked through a spreadsheet that collects the data for later processing. The data is obtained by the operator and the LST system that provides information about the routes.

The operator is responsible to record the following information:

1.  Route: route of a truck that comes from a supplier and periodically supplies the same material.
2.  Entry time: the time the truck enters the unloading tracks.

3. Exit time: the time the truck leaves the unloading tracks.
4. Incidents: the record of any incident that involves a loss of time during the download process.

The LST system will provide the following information:

1. Time window: the planned time interval of one hour during which the truck must arrive.
2. Unloading points: different areas of the plant where the truck must be unloaded, depending on the material being transported.
3. Supplier: the company that supplies the material.

Once the data is recorded in the spreadsheet, the following information is calculated:

1. Gross time: total time that the unloading track has been occupied by a truck.
2. Net time: time that the unloading track has been occupied by a truck without considering the break time and the snack break.
3. Unloading time: actual time spent unloading the truck.
4. Punctuality: it is considered punctual if the truck enters the unloading track in the planned time window.

In addition to the data described above, the following information must be recorded:

1. Shift: For night shift TN, TA, and TB for morning and afternoon shifts, respectively, which are rotated.
2. Day: the day the download is made.
3. Week: number of the week according to the calendar.

Table 3 presents a sample of the information collected from the process. It shows an abstract of the three shifts, the day of the week, week number, route code, time window, unloading points, entry time, exit time, gross time, net time, unloading time, and punctuality. Some of the information was modified for company confidentiality.

**Table 3.** The collected information from the unloading process to calculate the indicator.

| N° | 86 | 87 | 88 | 107 | 108 | 109 | 117 | 118 | 119 |
|---|---|---|---|---|---|---|---|---|---|
| Shift | TN | TN | TN | TA | TA | TA | TB | TB | TB |
| Date | 2/13/2020 | 2/13/2020 | 2/13/2020 | 2/13/2020 | 2/13/2020 | 2/13/2020 | 2/13/2020 | 2/13/2020 | 2/13/2020 |
| Week | 7 | 7 | 7 | 7 | 7 | 7 | 7 | 7 | 7 |
| Route | MB23X | PTC3X | PT63Y | MB14H | SK14M | PT24L | E624R | Delivery | E334N |
| Time window 1 | 21:00 | 21:00 | 22:00 | 10:00 | 10:00 | 10:00 | 15:00 | - | 14:00 |
| Time window 2 | 22:00 | 22:00 | 23:00 | 11:00 | 11:00 | 11:00 | 16:00 | - | 15:00 |
| Unloading points | D51E | D51E | D51E | RHE | D51E | D51E | D51E | | D51E |
| Unloading points | | | | D51E | | | | | |
| Entry time | 22:50 | 23:15 | 23:40 | 10:40 | 11:05 | 11:30 | 15:25 | 16:00 | 16:15 |
| Exit time | 23:10 | 23:30 | 23:55 | 11:10 | 11:30 | 12:25 | 16:00 | 16:15 | 16:35 |
| Gross time | 0:20:00 | 0:15:00 | 0:15:00 | 0:30:00 | 0:25:00 | 0:55:00 | 0:35:00 | 0:15:00 | 0:20:00 |
| Net time | 0:20:00 | 0:15:00 | 0:15:00 | 0:30:00 | 0:25:00 | 0:55:00 | 0:35:00 | 0:15:00 | 0:20:00 |
| Unloading time | 0:20:00 | 0:15:00 | 0:15:00 | 0:15:00 | 0:20:00 | 0:55:00 | 0:35:00 | 0:15:00 | 0:20:00 |
| Supplier | ABC | DEF | GHI | JKL | MNO | PQR | STU | | WVX |
| Punctuality | NOK | NOK | NOK | OK | NOK | NOK | OK | | NOK |

Time spent in the data collection phase was five months. During this period, approximately 2135 trucks unloading were recorded. In some periods and shifts, no data were reflected because the company was closed due to the health crisis caused by COVID-19.

The OPE indicator was calculated daily by shift and each factor was studied separately. Each factor was calculated with the equations indicated above and considering the losses established for each one of them. The total time of the shift for the availability calculation was 7 h and 42 min, and only 18 min break for the snack was considered. The ideal time for the performance calculation was considered as the average unloading time multiplied by the number of trucks served during the shift. Figure 5 shows the average rate of the four

factors per shift and week. The shifts are divided into morning (MO), afternoon (AF), and night (TN).

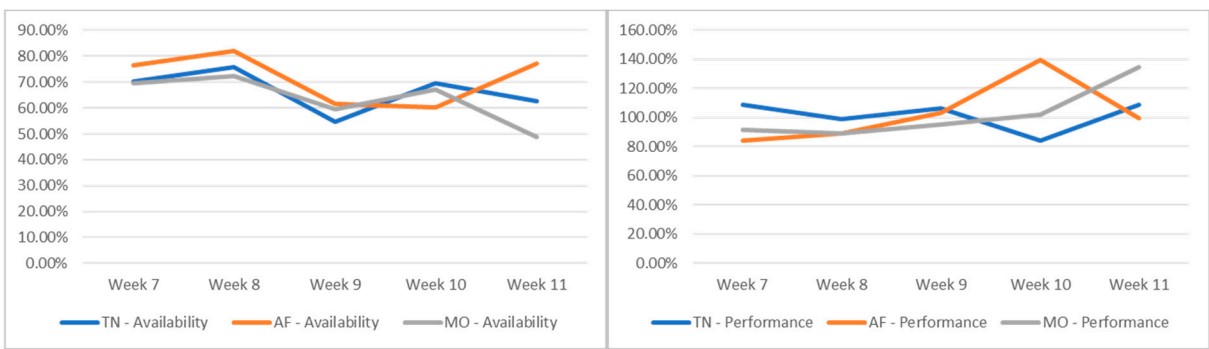

a) Average availability rate per week      b) Average performance rate per week

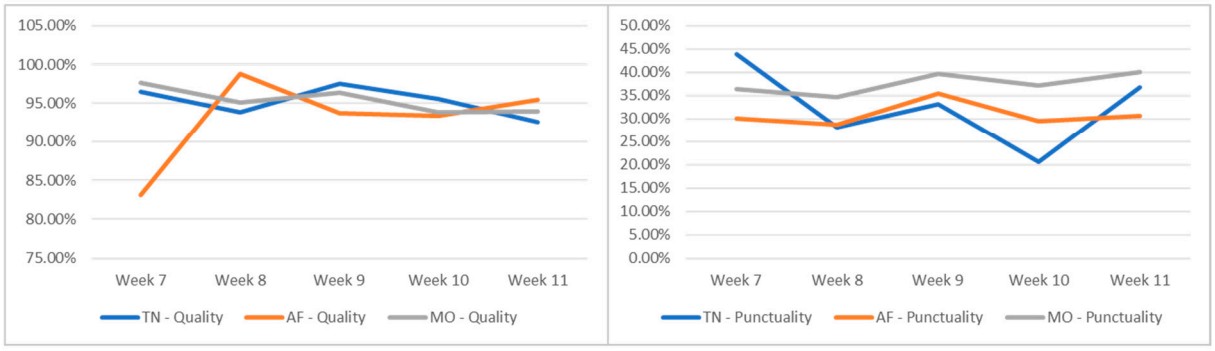

c) Average quality rate per week      d) Average punctuality rate per week

**Figure 5.** The average rate of the four factors per shift and week (**a**) availability; (**b**) performance; (**c**) quality; (**d**) punctuality.

The factors with the lowest averages that affect process performances are punctuality and availability. These two factors are the most influential in decreasing the final OPE result regardless of the shift. Availability is around 50 to 70%, being higher in the afternoon shift most of the time. The results indicated that the greatest loss of time is due to the lack of trucks to attend. The punctuality factor was added as an important part of the indicator, values between 25 and 35% were presented. Analyzing this result in-depth, the company detected that more than 50% of the trucks arrive outside their assigned time window. Most of the time, the delay is due to traffic jams, carrier outsourcing, and supplier material availability. The performance and quality factors are very close to the desired value. The performance on certain occasions provided values above 100%. After some meetings with the management team, it is notable that the average ideal time may be the cause of performance results, which will be evaluated in the future.

Figure 6a shows the OEE calculation (availability * performance * quality) and Figure 6b shows the OPE (availability * performance * quality * punctuality) during June 2020. In the OEE the morning and afternoon shifts maintain a similar behavior, the average is around 58%, whereas the night shift maintains a lower performance. The OPE presents great variations in its results attributed mainly to the punctuality factor. Results demonstrate that with the punctuality factor from morning and afternoon shifts, OPE varies on average around 23%. During this month, the night shift was irregular and was only worked from Monday to Friday, due to the restrictions of the health crisis for both cases. Figure 6c presents a comparison between the OEE and OPE of the different work shifts. It can be highlighted how the punctuality factor decreases the performance of the unloading process.

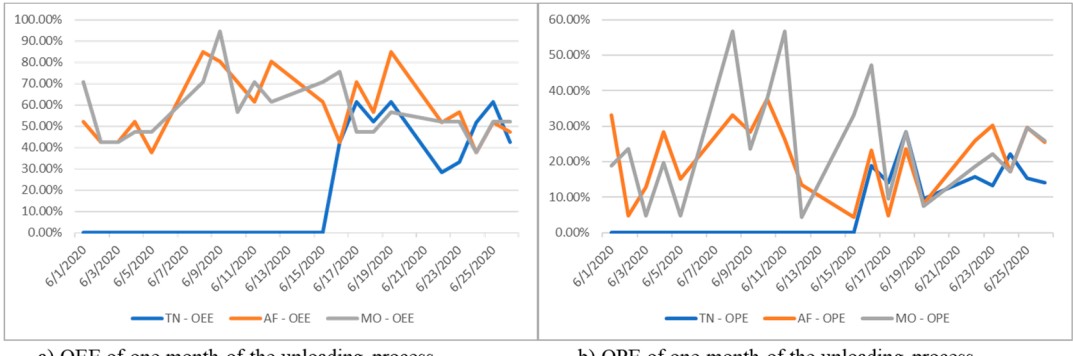

a) OEE of one month of the unloading process      b) OPE of one month of the unloading process

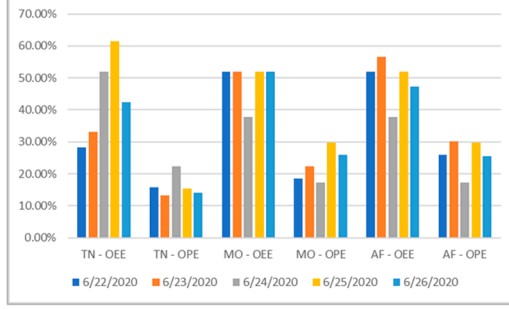

c) Unloading process OEE vs OPE

**Figure 6.** Graphic results (**a**) OEE of one month; (**b**) OPE of one month; (**c**) One week of OEE vs. OPE.

All these results were analyzed in multiple meetings with the management team, originating initiatives of improvements in the unloading process. The successful implementation of the OPE was due to a critical collaborative inquiry, where there was learning through action and reflection.

The proposed framework enables collaborative participation within the AR methodology used in this study, motivating a cultural change. Moreover, it was favorable for the company because it promotes participation from different organizational levels that conformed the management team in all the AR phases. The continuous improvement cycle that our framework focuses on is the result of the cohesion and synergy of the teamwork.

## 6. Discussion

This framework contributed to data digitization to encourage faster and more transparent results. Industry 4.0 and its pillars are tools that help the evolution of management systems [24]. This framework can lay the bases for other organizations to digitally transition their processes as part of a continuous improvement cycle [50]. The continuous improvement cycle is fed by KPIs information to eliminate activities that do not add value to a process. The constant evolution of production processes forces the organization to look for data management options to make data-driven decisions rather than intuition.

Through the framework proposed it can be possible to record and measure the process to obtain data for continuous improvement. Besides, the OPE allows the losses identification to take action and generate continuous improvement for lean performance.

The success of this collaborative research provides a practical contribution to the company that will improve the unloading process with the OPE implementation. Besides, it generates theoretical knowledge that can be applied to other departments or companies.

Once the AR cycle ended, we can provide answers to the three initial research questions.

1.  OPE indicator results do measure the effectiveness of the lateral unloading process. The results indicated that the categorized variables with the project team provided information on the four factors of the OPE and their influence on the unloading process.
2.  The AR methodology provided solutions related to the unloading process procedure and documentation. It was possible to record and organize the information

to calculate the performance indicator. Moreover, through the synergy between the researchers and the management team, it was possible to reduce bias in data collection and analysis of the results. Bias is reduced by the participation of external and internal personnel, improving the objectivity of the information analyzed. Through the collaborative participation of the researchers and the practitioners during the study, the new OEE adaptation could be measured and validated.

3. The study identified several opportunities for improvement, which are detailed in the theoretical and practical contributions.

### 6.1. Theoretical Contributions

Some authors consider the original OEE insufficient [28], as it was only used to measure the effectiveness of particular equipment. Many manufacturers have customized OEE to fit their particular industrial requirements [8]. This study makes two strong contributions. The first contribution is the adaptation of the OEE indicator to other domains, in this case to a logistic process. The modification contributes to the identification of the three OEE parameters (availability, performance, and quality) and punctuality as an external factor that is considered important in the process. It was possible to measure the actual operating time during the unloading process with the availability factor. The performance allowed us to calculate the ratio of the standard time vs. the real-time of the unloading process. The quality factor was measured based on the number of trucks that fulfill cargo requirements for unloading conditions among the total number of trucks unloaded in the shift. Finally, with the punctuality factor, we can identify the number of trucks that arrive within their time window. The punctuality factor measures the goods on-time delivery from the suppliers, one of the most important performance variables in a logistic process. [47].

The second contribution is to increase scientific knowledge in the logistics sector through the AR methodology [15]. Closing the gap between theory and practice [16] by the synergy between the company and the researchers in developing AR to solve an organizational issue. Through this AR, self-reflection of the team members was achieved to obtain improvements in the unloading process. The collaboration among members for problem-posing and answering questions for decision-making was crucial. The use of the AR methodology during the research contributed to a practical transformation within the company and the advancement of theoretical knowledge. Data gathering and analysis were accomplished by the active participation of the researchers during approximately seven months and the involvement of the management team in the implementation of the framework.

The collaboration between researchers and practitioners was valuable. It was possible to generate knowledge for the creation of the new indicator by the researchers, which was then validated by the management team of the company. This new metric allows to measure the unloading process and generate continuous improvement plans tailored to organizational needs, increasing their market competitiveness.

The ability to reach a high level of practical relevance of business research is one of the major advantages of this method [14]. The effectiveness of the proposed method has been substantiated here. To increase the scientific rigor of these results, different points of view with the triangulation were achieved, reducing the bias during the study, and through the collaboration with the management team, we had access to reliable information.

### 6.2. Practical Contributions

To the best of our knowledge, this is the first time that the OEE has been adapted to measure the effectiveness of an unloading process. We believe that this framework can be replicated in other companies that perform loading or unloading of goods. This new metric for measuring process effectiveness provided a breakdown of losses that occur during truck unloading. It was possible to identify the factors that affect the process, many of which were attributed to the internal situation, but others were noted to be external to the company, these external factors unsetting the planning and unloading time of a truck.

Through the AR methodology applied in this research, it was possible firstly, to strengthen the relationship between the practitioners and researchers providing scientific knowledge to the field and practical knowledge for an organizational issue. This practical knowledge allowed for the identification of a set of best practices to increase market competitiveness. Secondly, the OPE became one of the performance measurement indicators of the organization, and the reflection meetings that were held to validate it were adopted by the organization.

At the organizational level were identified several improvements starting with the developed framework that can guide the application in other processes. Moreover, data collection that was held with several templates and registered in specialized spreadsheets improves the way organizations analyze process information. The framework allows the company to build up a transition to a data-driven management approach.

With the OPE results, the company identified two improvements that help resource efficiency. First, it was noticed that more than 50% of trucks do not arrive in the planned time window, affecting the unloading schedule by shifts. Improvements are being made in the planning of truck arrivals, and meetings have also been held with suppliers to take joint action. Establishing a collaborative process with the supplier will strengthen the process flow between them [48]. The second improvement is the time redistribution of the operator to the unloading process. It was identified that the operator has a lot of leisure time with the availability parameter, so it was decided that the operator would attend two unloading areas.

Also, as a consequence of the self-reflections, a reevaluation of the calculation of the theoretical or ideal unloading time was being considered. This seeks to improve the presented results of the performance parameter above 100% on some occasions. A more accurate result could be obtained by calculating the average unloading time per route.

After a critical discussion with the management team, we defined the ideal OPE range for the company in the unloading process as between 40 and 50%. We want to highlight that this range considers that OPE results are influenced by external and internal factors. Some factors are not directly controlled by the company and cannot be compared with the OEE ranges obtained in production. Few authors proposed ideal ranges of OEE, establishing between 30 to 80% [29] and worldwide studies in manufacturing plants estimate an average of 60% [32]. The company continues evaluating improvement options with the OPE results. A permanent committee was established after the first OPE results to reflect on the necessary improvements to the unloading process, to achieve a higher percentage of the indicator.

This proposed framework is the first approximation to be developed for other logistics processes or equipment.

## 7. Conclusions

The OEE has been used as an indicator to control and monitor productivity and improve performance [49]. In this paper, a new indicator OPE has been introduced as a measure of a logistic process. Besides, the availability, performance, and quality factors considered in the original OEE, the punctuality factor is included. Punctuality was considered a determining factor to avoid bottlenecks in the process. The indicator helps to identify losses that occur during the process, intending to implement improvements that help the company's competitiveness.

This study was done with a collaboration between researchers and practitioners. AR methodology used considers the spiral cycle of plan, act, observe, reflect, and then reviews the process again.

First, the problem was established as the parameters and scope of the project were identified. Data from different sources were collected, and the indicator was calculated and analyzed.

Finally, theoretical and practical contributions could be evidenced in the process. OEE was implemented in a different domain than production. Losses identification that occurs during the lateral unloading truck process was imperative for the adapted OEE. Improve-

ments in the process were identified to increase work efficiencies, such as action plans to regulate the timing of the truck's arrival and distribution of the operator availability in the unloading areas. The AR methodology is flexible, and it can be applied in various industry domains since it involves the participation of multidisciplinary teams and promotes organizational changes. With this collaboration in action between researchers and practitioners, it was possible to generate and validate theoretical knowledge applied in the field. The AR provides a practical transformation and contributed to the advancement of scientific knowledge. It also encourages active and collaborative participation where a systematic learning and reflective process is developed throughout the action research cycle.

The study is not devoid of limitations. Some of the limitations that emerged in the application of the OPE were the global health crisis, which affected the company's operations and data collection. Other limitations were the study of only one unloading area. The company has several unloading areas, but for this research, the largest one was taken, which has two unloading tracks. Finally, for further research, the authors consider this indicator can be applied to the loading process with slight modifications. Moreover, it could be applied to other types of organization and cargo.

**Author Contributions:** Conceptualization, M.P.L. and J.R.; Methodology, M.E.H.K. and L.d.C.N.C.; Validation, L.d.C.N.C.; Formal analysis, L.d.C.N.C.; Investigation, L.d.C.N.C.; Data curation, M.E.H.K. and L.d.C.N.C.; Writing—original draft preparation, L.d.C.N.C.; Writing—review and editing, M.P.L., M.E.H.K., P.M. and L.d.C.N.C.; Supervision, M.P.L. and J.R.; Project administration, J.R. and J.C.S.C. All authors have read and agreed to the published version of the manuscript.

**Funding:** This research received no external funding.

**Institutional Review Board Statement:** Not applicable.

**Informed Consent Statement:** Not applicable.

**Data Availability Statement:** The data presented in this study are available on request from the corresponding author. The data are not publicly available due to the privacy policy of the organization.

**Acknowledgments:** The authors would like to acknowledge the scholarship granted by the government of Panama under the IFARHU—SENACYT program for a Ph.D. candidate from 2018 to 2021.

**Conflicts of Interest:** The authors declare no conflict of interest.

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
