# Peer review of "Developing and Implementing a Lean Performance Indicator: Overall Process Effectiveness to Measure the Effectiveness in an Operation Process"

_machines, doi:10.3390/machines10020133_

Round 1

Reviewer 1 Report

The aim of the manuscript entitled “Integrating a lean performance indicator to measure the efficiency in an operation process” is to implement a procedure to record, measure, and study the process of lateral unloading trucks. The manuscript is interesting in the practical domain since the OEE has been adapted to measure the effectiveness of an unloading process by analyzing losses and eliminating non-value-adding activities.

However, the manuscript needs to be improved in the following aspects:

In the introduction section, the authors should provide the research motivation.

The manuscript is not presented in a well-structured manner. The section Methodology and case description was presented before theoretical background. The theoretical background should be presented within the Introduction section. The authors describe the performance measurement system in general. They should analyze the current methodology for measurement of efficiency of the logistics process instead.

Since the Special Issue focuses on advances in applications of lean manufacturing methodologies, the results of the proposed research should be also discussed in that way.

The study was performed in one company, in only one unloading area. Is the proposed methodology applicable in other areas of the industry? What are the limitations?

There are a significant number of cited references that are not current. The authors should use references that have been published mostly within the last 5 years.

Author Response

Dear Reviewer,

Regards

Reviewer 2 Report

In this paper, the Authors try to build a new indicator derived from OEE, named Overall Process Effectiveness (OPE), with the aim to measure the efficiency of an operation process. The article and the approach proposed are interesting, but there are several issues to be addressed, especially regarding the abstract and the methodological procedures used to conduct the research. This is essential in a research paper.

The abstract needs to focus on objective, methodology, key findings, and research contributions. These contributions are not properly explored in the abstract.

In Section 2, there are also explanations that should be presented in the framework section (Section 4), in which authors should explain how they conducted the research, how they developed the proposed approach. I suggest to shift Section 2.1 after Section 3. In addition, in Section 2, there many paragraphs that regard the case study. Please, remove these sentences and rewrite Section 2.

It is stated (Abstract) that “The framework was developed with the researchers’ and practitioners’ experiences, the data was collected and analyzed; some improvements were applied and finally, a critical 18 reflection of the process was done…” Reading the manuscript, only two improvements were suggested. Is it correct? In my opinion, only two improvement actions are very low to implement benefits in the truck unloading process. Please help me clarify these aspects. 

Finally, I ask you if the OPE has been recalculated after the implementation of corrective actions, in order to see the improvements between the AS-IS and TO-BE status.

Author Response

Dear Reviewer,

Regards

Reviewer 3 Report

The manuscript is about proposing a lean performance indicator, named OPE – overall process effectiveness, derived from OEE, and developed and validated through collaborative participation. The work is interesting and relevant, with a sound research approach including a practical implementation case. Nevertheless, there are several aspects requiring major changes.

  • The title does not match with the work developed and it is vague. It should be concise and include the name of the proposed indicator, and it should reflect the results/outcomes
  • There is a difference between efficiency and effectiveness. In fact, OEE does not measure efficiency, but equipment effectiveness: how effective the equipment’s time is used. Efficiency is about the use of resources, and time is not a resource. So, the authors should distinguish the two concepts from the very beginning of the document (abstract included), and use them properly along with the manuscript. About the OPE components, please refer to the next points.
  • The abstract is concise, just the efficiency/effectiveness imprecision should be resolved
  • Introduction: Must be reorganized. In line 33 a comment and is difficult to relate it with transport “Previous research has established through interviews with practitioners that there are two types of value-added activities, transportation, and unloading/loading activities”. The use of expressions like “To the best of our knowledge,…” is not scientifically sound. The authors should refer to a survey in the most prominent data basis using the OEE term as keyword and by analyzing the results retrieve conclusions about the novelty of the proposed research. This is missing. And there is some incoherence with lines 46 to 48 of the Introduction.
  • This phrase in line 41 of Introduction has several mistakes “OEE indicator is used to measure the efficiency level by analyzing losses and eliminating non-value-adding activities.”: a) OEE is about effectiveness, not efficiency; b) OEE does not eliminate NVA activities, just identify how the equipment time is used; c) the NVA identification must be done by an analytical work, like 5Whys, to identify why availability or performance is being wasted (e.g lower speed is not an NVA, and can be identified by OEE).
  • There is incoherence also between lines 51 and 52 of the Introduction and the lines 13 and 14 of the abstract: two aims for the paper.
  • The contribution of the paper and the novelty is not clear at all in the Introduction, and it is not consistent with the Abstract.
  • About section 2, reference is missing in Figure 1 , as well as text in lines 81 to 84.
  • The paragraph starting in line 85 is confusing. It applies AR model to the case but the case was not properly presented. An overall presentation of the case study is missing in the start of section 2. Also the research questions are out of context since the new indicator was not introduced or the case study. Again efficiency is used not in the correct way.
  • So, Section 2 needs to be reformulated to have a logic and sequential logic: 1) the aspects presented; 2) before 2.1 called case description (line 160), the case is presented line 111 to 159.
  • Section 3 should be shortened regarding OEE explanation and content – is too descriptive. It should just focus on the relevant aspects and use KPI-related papers to express the importance of indicators and identify what is missing. Examples of papers on KPI:
    • https://doi.org/10.1007/s11135-013-9945-y
    • https://doi.org/10.1016/j.fcij.2016.04.001
    • https://doi.org/10.3390/app10186469
    • https://doi.org/10.4018/978-1-5225-5445-5.ch010
  • Section 4 should be a consequence of the gaps found in section 4, presenting the new indicator. So, in fact, sections 3 and 4 should be after section 1, and section 3 should be after sections 3 and 4, and there the research question would make sense.
  • Section 5 should include more reflections of one important stakeholder: the company elements involved in the study
  • In the discussion, section 6, considerations should be done regarding the present trend of digital transition in production management and Continuous improvement/kaizen, so papers like the following ones should be used to align the finding with data-driven management (potential of this indicator to be used in an IoT environment):
    • https://doi.org/10.3390/su11164291
    • https://doi.org/10.3390/app11167671
    • https://doi.org/10.3390/app11167648
  • The conclusions section should be numbered as 7.

Author Response

Dear Reviewer,

Regards

Round 2

Reviewer 1 Report

The authors have improved their manuscript according to previous comments.

Author Response

Dear Reviewer,

We appreciate your experience and the time you spent reviewing our work.

Regards,

Reviewer 2 Report

The Authors carefully revised the manuscript.
My integrations have been implemented. If they have also corrected the other Reviewer's addition, the manuscript can be published.

Author Response

(The authors gave the same response as above.)

Reviewer 3 Report

The manuscript is about proposing a lean performance indicator, named OPE – overall process effectiveness, derived from OEE, and developed and validated through collaborative participation. The work is interesting and relevant, with a sound research approach including a practical implementation case. The work was improved, mainly in

  • Title was improved.
  • The use of efficiency and effectiveness was improved.
  • Abstract and Introduction were really improved.
  • A new section dedicated to Literature review was introduced.
  • Other aspects were improved.

Nevertheless, some aspects need to be improved

  • English quality must be improved (concordance and verbs times).
  • Several additional papers were suggested to:
    1. Give a more updated discussion about knowledge gaps regarding KPI
    2. Regarding the present trend of digital transition in production management and Continuous improvement/kaizen
    3. And no modifications were done regarding these 2 aspects. Here are examples of papers that should be considered to discuss and frame the innovations proposed
      1. https://doi.org/10.1007/s11135-013-9945-y
      2. https://doi.org/10.1016/j.fcij.2016.04.001
      3. https://doi.org/10.3390/app10186469
      4. https://doi.org/10.4018/978-1-5225-5445-5.ch010
      5. https://doi.org/10.3390/su11164291
      6. https://doi.org/10.3390/app11167671
      7. https://doi.org/10.3390/app11167648
  • I do not agree with the authors “We review the proposed paper and consider the OEE literature review in our study had the minimum knowledge base to understand where our approach comes from.” Is relevant to explain the importance of the innovation proposed regarding recent papers on KPI and also regarding its readiness with the digital transition.

I recommend major revisions because a better scientific framing of the proposed innovation is mandatory, but it is just that.

Author Response

Dear Reviewer,

Regards

Round 3

Reviewer 3 Report

ready to be accepted